# Bioabsorbable Composites Based on Polymeric Matrix (PLA and PCL) Reinforced with Magnesium (Mg) for Use in Bone Regeneration Therapy: Physicochemical Properties and Biological Evaluation

**DOI:** 10.3390/polym15244667

**Published:** 2023-12-11

**Authors:** Rubén García-Sobrino, Marta Muñoz, Elías Rodríguez-Jara, Joaquín Rams, Belén Torres, Sandra C. Cifuentes

**Affiliations:** 1Department of Applied Mathematics, Materials Science and Engineering and Electronic Technology, Universidad Rey Juan Carlos, Calle Tulipán s/n, 28933 Móstoles, Spain; marta.munoz@urjc.es (M.M.); joaquin.rams@urjc.es (J.R.); belen.torres@urjc.es (B.T.); 2Instituto de Cerámica y Vidrio, Consejo Superior de Investigaciones Científicas (CSIC), Campus de Cantoblanco, c/Kelsen 5, 28049 Madrid, Spain; eliasrj@icv.csic.es

**Keywords:** bone regeneration, bioabsorbable materials, composites, PLA, PCL, magnesium

## Abstract

Improvements in Tissue Engineering and Regenerative Medicine (TERM)–type technologies have allowed the development of specific materials that, together with a better understanding of bone tissue structure, have provided new pathways to obtain biomaterials for bone tissue regeneration. In this manuscript, bioabsorbable materials are presented as emerging materials in tissue engineering therapies related to bone lesions because of their ability to degrade in physiological environments while the regeneration process is completed. This comprehensive review aims to explore the studies, published since its inception (2010s) to the present, on bioabsorbable composite materials based on PLA and PCL polymeric matrix reinforced with Mg, which is also bioabsorbable and has recognized osteoinductive capacity. The research collected in the literature reveals studies based on different manufacturing and dispersion processes of the reinforcement as well as the physicochemical analysis and corresponding biological evaluation to know the osteoinductive capacity of the proposed PLA/Mg and PCL/Mg composites. In short, this review shows the potential of these composite materials and serves as a guide for those interested in bioabsorbable materials applied in bone tissue engineering.

## 1. Introduction

Economic, technological and demographic difficulties associated with conventional therapies such as organ transplantation, implants and/or surgeries have generated worldwide interest over the years in research and clinical applications in the field of Tissue Engineering and Regenerative Medicine (TERM) [1]. Over the last decades, this sector has become an emerging and multidisciplinary field that seeks the maintenance, improvement and/or reconstruction of damaged tissues or organs from the synergy formed between different disciplines such as materials science, cell biology and biochemistry [2,3]. In this way, the search for and optimization of new materials with improved performance is, therefore, of great importance in the TERM sector to address increasingly complex pathologies.

In the case of bone-type tissues, although the native bone presents intrinsic regeneration capacities after damage or fracture processes, there are lesions that do not regenerate in a natural way. These defects generally are greater than 2 cm in some anatomical points and are associated with degenerative diseases or include congenital defects related with tumor removal processes [4,5]. As a solution, surgeries incorporating implants are performed, which in the cases of elderly patients may involve high risks to their health [6]. Improvements in TERM-type technologies have allowed the development of specific materials that, together with a better understanding of the biology of bone tissue structure, have provided new opportunities for bone regeneration with in vitro [7,8] or in vivo models [9,10] and even in the clinical phase [11,12].

The journal “Biomaterials” defines the concept of biomaterial as a material designed so that, individually or as part of a system, it can direct, through interactions with living tissue, the course of any therapeutic procedure [13]. These materials, linked to the sector, have gone from being characterized as simple materials that do not interact with the original tissue (bioinert) to bioactive materials that, in addition to generating a harmonious interface with the area to be replaced, should stimulate the processes of cell adhesion, proliferation and differentiation [14]. Thus, in the continuous search for possible improvements in patients’ life, biomaterials have revolutionized modern medicine with a wide range of applications in the biomedical sector, including their use as implants (dental implants, heart valves, intraocular lenses, ligaments, vascular grafts, etc.) or equipment for medical devices (artificial hearts, biosensors, pacemakers, etc.) [15,16]. 

Bioabsorbable materials, protagonists of this work, are temporary structures that breakdown and are cleared from the physiological environment. These materials are designed to be non-toxic and gradually degrade, allowing mechanical stresses to be transferred gradually to the healing tissue [17]. This quality makes them even more interesting for certain applications such as research in the TERM sector, since in cases such as surgical treatments, they avoid second surgery processes thanks to their degradation [18]. That is why the study of bioabsorbable materials for implants is essential at this moment; in fact, the global market for this type of implants is expected to grow annually at 10.5% between 2022 and 2030 [19].

Depending on the family of the materials used, these elements can be metals, which include magnesium (Mg), zinc (Zn) and iron (Fe) [20]; ceramic, mainly bio glasses and calcium phosphate [21,22,23]; or polymeric type. The last one can be divided according to its origin into natural polymer (collagen or cellulose) and/or synthetic (commonly based on poly glutamic acid (PGA), poly lactic acid (PLA), poly caprolactone (PCL), poly (lactid-co-glycolid) (PLGA), polyethylene terephthalate (PET) or poly urethane (PU)) [20,24]. Polymers have traditionally been the most widely used family of bioabsorbable materials, with an approximate percentage of 81.4% compared to the 18.6% represented by the use of bioabsorbable metals [25]. Most commercially available resorbable implants are based on polymers and are applied in low-load bearing applications such as ankle, knee and hand surgery or cranio-maxillofacial surgery [26]. Global manufacturers of implants are committed to the growth of bioresorbable implants’ market. Efforts are focused into expanding the range of applications (for instance, the treatment of long-bone fractures or critical size defects) and offering more comfort to the patient and the surgeon by improving the performance of the implants. These goals necessarily imply the improvement of bioresorbable materials. A fourth family of materials, the so-called composite materials, has generated special interest in the bioabsorbable field in recent years. The properties resulting from the combination of materials from different families give rise to hybrid systems that allow to obtain a composite with superior performance than each material separately. Given their characteristics and interactions in the organism, polymers are ideal matrices for the study of composite biomaterials due to their versatility in terms of geometry, morphology, microstructures and synthesis, making them systems that can perfectly mimic the original structure of an organ [27,28].

In reference to studies related to bone lines, in addition to elements that favor osteoinductive processes, the inferior mechanical properties of polymers with respect to bone make it necessary to study reinforcement processes that allow obtaining hybrid materials with optimal density and mechanical properties [29]. This mechanical compatibility requirement is highly important. Thus, significantly different elastic modulus values with respect to the host bone tissue can lead to patient problems such as osteoporosis and osteolysis (moduli above and below the native value, respectively) [30]. In short, the synergy between the polymer matrix and reinforcement in composite materials makes it possible to achieve hybrid characteristics that match the properties required for the proper reconstruction of damaged tissues [31].

Regarding the study of composite material constructions with polymeric matrix and bioabsorbable capacity, in the last decades, mainly ceramic reinforcements have been extensively studied [32], such as collagen/hydroxyapatite [33], cellulose/calcium phosphates [34], PLA/hydroxyapatite [35], among others. The study on this kind of reinforcement is due to their compositional similarity to bone tissue and high absorption rate. In addition, acceptable mechanical and physical properties such as high hardness and wear motivate their use. However, their high brittleness limits their use in certain applications when high loads are required [36]. To overcome these limitations, Mg biodegradable metal is presented as an alternative solution to ceramic reinforcements. Mg is characterized by a high osteoinductive character due in part to its high presence in tissues of a bony nature (50% of the Mg content in the human body is found in this type of tissue). Also, it is necessary to mention that Mg has an elastic modulus similar to that of native osseous tissue (3–20 GPa) [37]. In the last decade, research on Mg as a biomaterial for bone regeneration has remarkably increased, and few commercially available biodegradable devices based on Mg have already been developed. These devices are mainly applied in low-load bearing applications [38].

The aim of this manuscript is to show an updated and comparative bibliographic review of bioabsorbable composite materials applied in bone regeneration based on the idea of combining a polymeric matrix with a bioabsorbable metal reinforcement. This review focuses on two of the most studied synthetic FDA-approved polymeric matrices (PLA and PCL), both reinforced with Mg alloys as the bioabsorbable and osteoinductor metal to improve their performance in bone-TERM studies. In this manuscript, the influence of the reinforcement on the physical properties and biological evaluation (in vitro and in vivo studies) has been addressed.

## 2. Polymers as Bioabsorbable Materials for Bone Tissue Engineering

### 2.1. Aliphatic Polyester as Bioabsorbable Material: PLA and PCL

Since the last century, research on polymers has mainly focused on healthcare applications (controlled drug release, tissue engineering, wound healing, etc.) [39]. Although it is not so often mentioned to the vast majority, this family of materials is a fundamental and inherent part of nature itself. As previously mentioned, depending on their origin, polymers can be of natural or synthetic origin. In the case of the first ones, they coexist in our daily life, since they involve protein macromolecules of our body to elements such as silk generated by certain animals, for example, silkworms or spiders. It is for this reason that the study of these materials, in relation to applications in the health field, is so important, given their inherent excellent behavior in terms of biocompatibility and bioabsorbable capacity, complementary to the fact that they lack immunogenic response. Unfortunately, for certain applications, natural polymers are limited by their low mechanical properties and their lack of versatility in terms of chemical modification, which limits their study options [40]. To compensate these limitations, polymers of synthetic nature are studied for their versatility in terms of functionalization and chemical modification that allows to obtain physicochemical and biological characteristics according to the desired requirement [41,42]. The synthetic combination between different monomeric units or chemical modification of different structures allows the optimization of the macromolecular chain according to the required application.

Among the synthetic polymers applied in TERM studies, aliphatic polyesters, a family of polymers such as PLA, PGA, PCL or PLGA, have been studied. Their bioabsorption is based on the hydrolysis of the ester groups of the polymer chain, so this behavior can be modulated with compositional modifications that alter the structure, molecular weight and composition [43,44]. In the case of this work, two of the most studied and mentioned bioabsorbable aliphatic polyesters, PLA and PCL, are subsequently described in more detail.

#### 2.1.1. Polylactic Acid (PLA)

PLA is the most researched and used bioabsorbable polymeric material in medical applications (bone fixation material, suture, drug delivery microspheres and tissue engineering) [45]. PLA-based materials were approved by the Food and Drug Administration (FDA) [46] for direct contact with biological fluids in the 1970s and are obtained from the polymerization process of the lactic acid monomer unit (LA or 2-hydroxypropionic acid, C_3_H_4_O_2,_ see structure in Figure 1) using catalysts and rigorous conditions (temperature, pressure and pH) in a long-time process [45]. This molecule is an organic acid of natural origin that in 90% of the cases is generated from the fermentation of sugars instead of chemistry synthesis processes [47]. PLA is a chiral molecule, in which two possible enantiomeric units, L- and D-lactic acid, co-exist. Thus, through the polymerization process (by polycondensation, ring-opening polymerization or direct methods such as azeotropic dehydration or enzymatic polymerization) [48], it is possible to obtain different macromolecule conformations depending on the proportion of the repeated units: on one hand, it is possible to form macrochains composed of PLLA and PDLA, both with a high degree of isotacticity, causing a high degree of crystallinity; and on the other hand, it is possible to form macrochains using a combination of D- and L-lactic acid, generating PDLLA macromolecules of atactic structure, thus reducing the crystallinity values [49,50]. This degree of tacticity is reduced with D-isomer values higher than 8–15%, making the polymer amorphous [51]. It is well known that the higher the degree of crystallinity, the better the mechanical properties of the system (see Table 1 where values of the mechanical properties of the possible options presented by the PLA possibilities are shown). Similarly, it is necessary to highlight how the transition temperature values collected, also in Table 1, show that PLA and its homologues are solid and rigid at room temperature, since this temperature is below the glass transition temperature (Tg) value. The properties of the final material are dependent on different variables such as the processing temperature, the cooling rate and the molecular weight of the system [52]. Some of the common PLA processing methods are drying and extrusion, injection molding, compression molding, electro spinning, 3D printing, etc. [49].

As for the associated degradation of this aliphatic polyester, it is related to the interaction of microorganisms such as fungi, bacteria and algae [53]. In the case of physiological environments, with the intention of study in the biomedical field as a bioabsorbable material, the oxidative, enzymatic and catalytic reactions related to the pH of the aqueous medium are the causes of the degradation process mentioned. In this scenario, water penetrates more easily into the amorphous phases of the structure, causing a decrease in the molecular weight value involving the cleavage of the macromolecule and with it the consequent loss of mechanical and mass properties. Numerous variables condition the speed of degradation of the system such as the presence of catalysts, additives, plasticizers or impurities [49]. It is necessary to emphasize that PLA is known to have high degradation periods; the reason for this is based on steric hindrances where the alkyl group of the chain hinders the continuity of the process [54]. Specifically, the high crystallinity structure of PLLA impairs this process in terms of degradation times; in addition, acidic by-products are generated during its degradation process that can provoke inflammatory responses in the body. To remedy this situation, the PDLLA macrochain option is synthesized to increase the amorphic part of the system, so that the degradation times are reduced [55].

#### 2.1.2. Polycaprolactone (PCL)

The second of the polymeric and bioabsorbable matrices to be described is PCL, which was also approved by the Food and Drug Administration (FDA) as a biocompatible element [56]. This polymer is characterized by being a polyester of synthetic origin and partially crystalline (generally up to 80% crystallinity) [57], whose glass transition temperature (Tg) and melting temperature (Tm) are around −60 and 60 °C, respectively. Its polymerization process develops from the ring opening of the cyclic monomeric unit ε-caprolactone (ε-CL, C_6_H_10_O_2_, see structure in Figure 1) [58]. PCL mechanical properties vary depending on the degree of crystallinity and the molecular weight of the chain. Thus, as a polymer of synthetic origin, depending on the processing of the material, its properties can be optimized to suit the purpose of the application (see different properties of PCL summarized in Table 1). Also, it is important to highlight some of the common processes for PCL processing in biomedical applications: solvent casting, electrospinning, phase separation techniques, 3D printing, compression, injection molding or extrusion [59].

In terms of degradation and in a generic manner, being also an aliphatic polyester, PCL in the presence of water undergoes hydrolysis with subsequent degradation of the soluble oligomer generated by the cells and microorganisms in the environment [60]. The PLA mechanism process begins with a reduction in the molecular weight of the macromolecule, which implies a breakdown of the chain structure that facilitates the process of metabolization by cells and/or microorganisms [61]. In fact, it is the decrease in molecular weight itself that precisely controls the hydrolysis of the polymer. Because of the hydrolysis process, there is a loss of mechanical properties based on cleavage of the macromolecule, which finally leads to the consequent loss of mass of the bioabsorbable material. These stages do not have a fixed duration and depend on the characteristics of the medium and the structure and configuration of PCL itself, i.e., high molecular weight values impair the degradation of the material, as expected. Furthermore, the architecture and porosity of the material have a direct effect on the degradation process where thinner and more porous matrices favor the hydrolysis process [57].

**Table 1 polymers-15-04667-t001:** Physical and mechanical properties of PCL and PLA bioabsorbable polymers. Adapted from Farah, S. et al. [49] and Bartnikowski, M et al. [57].

Material/Properties	PLA	PLLA	PDLLA	PCL
Mw(×10^3^ g/mol)	120–800	60–800	120–270	50.4–124
Density (g/cm^3^)	1.21–1.25	1.24–1.3	1.25–1.27	1.11–1.46
Glass transition temperature, Tg (°C)	45–60	55–65	50–60	−60
Melting point temperature, Tm (°C)	150–162	170–200	-	56–65
Young’s modulus, E (MPa)	350–3500	2700–4140	1000–3450	252–440
Tensile strength, (MPa)	16.8–48	40–66.8	22.1–39.4	10.5–27.3
Strain at break; (ε_b_—%)	2.5–6	3–10	1.5–20	80–800

-: sample without melting point temperature. Amorphous structure.

## 3. Metals as Bioabsorbable Materials: Magnesium (Mg)

Traditionally, the good mechanical properties that characterize biomaterials of metallic nature have allowed them to be highly valued options in the selection of materials for biomedical applications related to bone lesions. Thus, for decades, their behavior towards the host tissue has been characterized as a bioactive relationship with it, as observed in the literature with numerous articles based on 316 L steels (also known as surgical steel), CoCr and titanium alloys such as Ti6Al4V [62,63,64]. In the last decades, the paradigm on the corrosion of metals has changed in the biomedical field, so that the so-called corrosion process is now referred to as a process of material degradation in a physiological environment, being ideal in biomedical applications where a bioresorbable device is appropriate [20]. Thanks to its biodegradable capacity, metals cited (Fe, Zn or Mg) play interesting roles as temporary orthopedic and vascular implants [65]. In the case of Fe, although its mechanical properties are similar to 316 L alloy and it being present in preclinical studies in the literature [66], Fe presents a very low degradation rate that is a problem for the bone regeneration process target; so, there is a continuous search for new iron alloys to accelerate its degradation rate [17]. On the other hand, in the case of Zn, although there are also studies in the preclinical phase [67], this metal has an intermediate corrosion rate between Fe and Mg and lower mechanical properties than Fe [17]. Finally, in the case of Mg, the biocompatible, osteoconductive co-protagonist of this manuscript, it presents values of elastic modulus similar to that of cortical bone and a degradation rate higher than that of Fe and Zn, making this biomaterial as the bioabsorbable metal par excellence at present [68].

Mg is a metal with a high presence in the human body, thus ensuring its biocompatibility. To be exact, it is the fourth trace element with the highest presence in the body, participating in many of the physiological functions including the absorption and metabolization of calcium itself in the bones [37]. The density value of this metal is close to that of cortical bone. Also, regarding mechanical capabilities, this metal in its pure state has elastic modulus values like those of bone, as mentioned previously (see Table 2). The symbiosis between low density and strength close to bone generates a value of specific strength, the results of which are interesting for tailoring properties [69]. 

Magnesium degradation in a physiological environment can be understood with Equations (1) and (2). It is known that the concentration of chloride ions in physiological media is high (150 mmol/L). Under these conditions, a thin layer of magnesium hydroxide is formed after the contact of magnesium with aqueous media. At this point, magnesium hydroxide reacts with the chloride anion to form magnesium chloride, soluble in water, and hydrogen in the gaseous state [51]. The rate of generation of hydrogen gas is the most common parameter to evaluate the corrosion of magnesium. As Song et al. [70] reports, for each ml of H_2_ generated, 1 mg of metal is diluted. The process of release of magnesium ions (Mg^2+^) associated with the procedure mentioned has been described as favoring the regeneration of bone tissue [71]. This process of regulation of osteogenesis does not seem to be entirely clear, but it has been demonstrated that the presence of Mg ions favors the growth and differentiation in processes of regeneration of this tissue [72,73].

Mg (s) + 2H_2_O (l) → Mg^2+^ (aq)+ 2(OH)^−^ (aq) + H_2_ (g)(1)

Mg(OH)_2_ (s) + 2Cl^−^ (aq) → MgCl_2_ (soluble) + 2(OH)^−^ (aq)(2)

Although pure Mg has already been applied in the pre and clinical phase based on the literature [74], this material shows some disadvantages for which improvements are required. For instance, it is necessary to design Mg alloys with better mechanical performance, lower degradation rate and easier conformability capabilities. In this sense, research groups have redirected their efforts towards alternative solutions such as the development and optimization of new Mg alloys, coating processes or modifying the microstructure to find suitable alloys for different applications in the biomedical field. In the case of Mg alloys, presence of other elements in the structure causes variations in the target properties; so, it is necessary to emphasize that even the presence of traces or impurities in low concentrations can cause noticeable variations in the final properties of the material. For example, it is described that Fe, Ni and Cu play a detrimental effect on Mg corrosion rate [75]. Aluminum (Al), the main magnesium alloying element, is characterized by a high solubility in magnesium (12.7 wt.%); thus, the higher mechanical capacity of aluminum itself improves the strength conditions of the base metal [76]. In addition, regarding corrosion, it is described that with 8 wt.% of Al, the corrosion resistance is improved thanks to the thin passive film formed of alumina (Al_2_O_3_). Another common alloying element Zn possesses a high solubility with respect to the base metal (6.2 wt.%). This situation also implies improvements in the mechanical properties of the material and in the properties related to corrosion. The reason for corrosion improvement is that in this case, Zn also generates a thin layer based on zinc oxide (ZnO) that slows down the advance of corrosion. In addition, it is also worth mentioning that Zn^2+^ ions have antibacterial and osteoinductive characteristics, ideal for bone biomedical applications [77,78]. Mg alloys containing both, Al and Zn, originally designed for the transportation industry where corrosion-resistant materials are needed, have been evaluated for temporary biomedical devices. The most studied are AZ31 (with 3 and 1 wt.% of Al and Zn, respectively), AZ61 (6% Al and 1% Zn) and AZ91 (9% Al and 1% Zn). These alloys have demonstrated to be biocompatible and to induce the formation of new bone by enhancing the mineralization process [79]. However, the potential toxicity of Al opens a debate within the scientific community as Al is known to be linked to dementia and Alzheimer’s disease [80]. It is for this reason that Alfree Mg alloys have been designed for biomedical applications. Finally, systems based on essential body elements such as Zn or Ca have demonstrated to have controllable degradation rates and good mechanical performance [81].

**Table 2 polymers-15-04667-t002:** Physical and mechanical properties of pure Mg and alloys compared with bone tissue properties [37,82].

Material/Properties	Cortical Bone	Trabecular Bone	Mg Pure	AZ91	AZ31	AZ61	Mg6Zn
Density (g/cm^3^)	1.8–2	1–1.4	1.74	1.81	1.77	1.80	1.84
Young’s modulus, E (GPa)	15–30	0.05–0.5	44	45	44.8	44.8	42.3
Tensile strength, m (MPa)	50–150	10–20	90	165–457	260	310	277–281

## 4. Bioabsorbable Composite Materials Based on Polymeric Matrix (PLA and PCL) Reinforced with Magnesium (Mg)

The motivation behind the combination of biodegradable polymeric matrices with magnesium is supported by the idea that a synergy between both materials can be obtained. The main hypotheses are that Mg could provide mechanical reinforcement and bioactivity enhancement of the polymer, and that the polymer in turn could control the corrosion rate of the metal. This review offers a compilation of the results achieved in the last years on bioabsorbable composite materials based on PLA and PCL reinforced with Mg. The parameters to be evaluated will include, on one hand, the behavior of the samples in mechanical studies, evaluation of degradation of the material in physiological environments and, on the other hand, cytocompatibility evaluation in in vitro and in vivo conditions. In this way, it will be possible to evaluate the main studies and compare the optimization processes and results obtained from the development of bioabsorbable hybrid materials applied in lesions of a bony nature.

### 4.1. Bioabsorbable Hybrid Materials Based on PLA Reinforced with Mg (PLA/Mg)

Composites based on PLA and reinforced with Mg have been designed and developed using different types of reinforcement and various fabrication processes. The final biological properties of a composite and its performance in vitro and in vivo depend on the nature of the constituents, the geometry and amount of the reinforcement, the matrix–reinforcement interface and the fabrication process. Multiple combinations are possible to create composites. For clarification purposes, studies based on PLA/Mg composites in this section are organized here in three categories considering the geometry of the reinforcement and the fabrication technology. In this way, the first category deals with studies where PLA has been reinforced with Mg particles, the second category includes PLA reinforced with Mg wires and the last one includes PLA/Mg composites fabricated using 3D printing technology.

#### 4.1.1. PLA Reinforced with Mg Particles

The study of PLA-based systems reinforced with Mg started in 2012 with the proof of the concept of the improvement of PLLA mechanical properties by incorporating 30 wt.% of Mg particles in the polymeric matrix. In the mentioned manuscript, Cifuentes, S. et al. [83] fabricated PLLA/Mg composites by solvent-casting and compression molding using Mg particles (<250 µm). PLLA/Mg showed higher compressive strength (101.3 MPa) and Young´s modulus (8 GPa) than pure PLLA (58.6 MPa and 2.86 GPa, respectively). Subsequent studies focused on developing PLA/Mg composites by different processing methods and on further improving the PLA mechanical performance by increasing Mg content. As a continuation of their research, Cifuentes, S. et al. [84] also explored the suitability of processing PLLA/Mg composites by hot extrusion. They homogeneously blended Mg particles (<50 µm) in different contents (0, 0.5, 1, 3, 5 and 7 wt.%) with a PLLA matrix (MFI: 35.8 g/10 min) using a Haake Minilab extruder. They found that Mg particles influenced PLLA thermal degradation during the extrusion process, and this led to a slight improvement in the stiffness of PLLA until 5 wt.% of Mg was achieved with a drastic drop of mechanical compressive performance in 7 wt.%. A later study on the extrusion of PLA/Mg composites achieved the incorporation of 15 wt.% Mg particles with a median particle size of 24.6 µm within polymeric matrices of PLLA (approximately 95,000 g/mol) and PDLLA (approx. 103,000 g/mol) [85]. They used a Rondol co-rotating twin-screw continuous extruder to compound the materials and incorporated Mg particles in contents of 0 as a control, 1, 5, 10 and 15 wt.%. Then, the mechanical properties of the composites were analyzed by Depth Sensing Indentation (DSI); an improvement in the modulus and hardness of composites was found when the reinforcement was higher than 5 wt.%. Mg reinforcement also influenced the creep behavior of polymeric matrices by incrementing the viscosity coefficient.

Based on the aforementioned studies, the amount of Mg particles that can be incorporated into the polymeric matrix depends on the manufacturing technology and the interaction between the matrix and the filler. In this sense, manufacturing PLA/Mg composites by injection molding [86] allowed the incorporation of less than 1 wt.% Mg particles into a PDLLA matrix (Figure 2a,b), while colloidal processing allows the incorporation of 50 wt.% Mg within the polymeric matrix (Figure 2c) [87]. The incorporation of low amounts of Mg in PDLLA by injection molding did not cause a strengthening effect in the polymeric matrix but induced beneficial effects in terms of mesenchymal stem cell (purified human bone marrow-derived MSCs) viability and macrophage responses. Authors suggested that small amounts of Mg in polymeric medical devices could promote osseointegration and reduce host inflammatory response [86]. On the other hand, incorporation of high contents of Mg particles in PLA did not necessarily imply a strengthening effect on the matrix.

Ferrández-Montero, A. et al. [88] modified the surface of spherical Mg particles using stabilizers, polyethylenimine (PEI) and cetyltrimethylenimine (CTAB). They created colloidal suspensions with a high Mg content in deionized water at pH 12, added the stabilizer and dispersed the suspension in PLA previously dissolved in tetrahydrofuran (THF). Colloidal mixtures with a PLA/Mg ratio up to 50/50 wt./wt. were casted in films where a homogeneous distribution of Mg particles was achieved by the improvement of the particle bond to the matrix. The mechanical properties of PLA/Mg films with different Mg contents (0, 5, 10, 30 and 50 wt.%) modified with PEI and CTAB have been studied using dynamic mechanical thermal analysis (DMTA). Authors found a better interaction between PEI-modified particles with PLA than CTAB-modified particles with PLA. The elastic modulus of the composite reinforced with 10 wt.% Mg modified with PEI was higher than that of PLA and the composite modified with CTAB with the same percentage of reinforcement. However, a significant loss of modulus was observed when Mg content increased to 30 and 50 wt.% with both surface modifications. Authors explained this behavior with the effect of Mg content on the crystallinity of the films. Mg particles interfered with the crystallization of the polymer during the casting tape process. The films with 10 wt.% Mg modified with PEI presented a crystallinity of 45%, while the crystalline fraction with 30 wt.% and 50 wt.% of filler decreased to 30% and 6% respectively. 

Regarding the in vitro degradation of PLA/Mg particles composites, Cifuentes, S. et al. [89] have conducted studies on the influence of Mg particle shape and the relevance of the nature and crystalline degree of the polymeric matrix [90]. The shape of Mg reinforcement played an important role in the degradation rate of PLA/Mg composites. Composites based on PLA (2002D) and reinforced with 10 wt.% Mg particles (less than 50 μm), which were spherical and irregular in shape (Mg-SPH and Mg-IRR, respectively), were fabricated and subjected to degradation studies in phosphate buffered saline (PBS) solution, at 37 °C after 7 and 28 days (Figure 2d). After these periods of time, their compressive strength was evaluated. Authors found that samples reinforced with irregular-shaped particles lost their physical integrity during the first week, while samples reinforced with spherical particles retained 96% of their compressive strength after 7 days and 60% after 28 days. In order to study the relevance of the matrix nature and crystallinity in the in vitro degradation of PLA/Mg composites, the manuscript of Cifuentes, S. et al. [90] compared composites based on two different polymeric matrices (PLLA and PDLLA) and the same matrix (PDLLA) with different crystalline fractions (amorphous and highly crystalline). Composites were reinforced with 10 wt.% of irregularly shaped Mg particles. The degradation rate of composites was followed by hydrogen release measurements. The authors found that the composite with PLLA matrix released hydrogen at a higher rate than that based on PDLLA. Regarding the effect of PLA crystalline degree, the composite with the high crystalline matrix degraded faster than the one with the amorphous matrix. The authors explained that the amorphous parts of the high crystalline polymer degrade faster than the amorphous polymer. Water penetrates preferentially into the amorphous domains in the high crystalline polymer.

The analysis of the effect of PLA matrix crystallinity on the cytocompatibility of PLA/Mg composites has also been studied [90]. Human bone marrow-derived mesenchymal stem cells (MSCs) have been cultured in DMEM medium for up to 14 days with amorphous and crystalline PDLLA and PDLLA/Mg composites with an amorphous and a crystalline matrix. It was found that cytocompatibility increased with the addition of Mg into amorphous PDLLA. However, cytocompatibility of composites with the high crystalline matrix was considerably lower than that of composites with an amorphous polymeric matrix.

On the other hand, Zhao, C. et al. [91] used the mouse osteoblastic MC3T3−E1 cell line to assess the effect of Mg content on the biocompatibility and biomineralization property of PLA/Mg composites. They fabricated composites based on PLA (1.24 g/cm^3^) and reinforced with 0 as the control and 2 and 5 wt.% of Mg particles (100 µm) by solvent casting and further compression molding. The viability of MC3T3−E1 cells was assessed by indirect tests and the biomineralization by direct tests. Cells were seeded in 24 h extracts prepared in alpha modified Eagle´s medium (α(MEM)) and incubated for 72 h. Authors reported no statistical difference between the cell viability in the extracts of PLA/Mg composites and that of PLA extracts. However, the average value of cell viability was higher for materials containing Mg (see Figure 3a for quantification and representative confocal microscopic images of cell staining). Also, the cellular calcium deposition marker was recorded for each sample after seeding the cells on each material for 14 and 28 days. Authors found a statistical higher number and area of mineralized nodules on samples containing Mg than on pure PLA. Nonetheless, different Mg contents (2% and 5%) did not show differences in mineralization. These results evidenced that the presence of Mg in PLA promoted bioactivity and osteoconductivity capacity of the polymer.

Lee, H. et al. [92] also studied the cell culture and proliferation capacity of the MC3T3−E1 cell line on PLA/Mg composites. They prepared the composites by mixing PLA (Ingeo 4032D) with Mg particles (average size of ~120 μm) in a high shear equipment at 180 °C. The polymeric matrix was reinforced with 0 as the control and 15 and 30 vol.% of particles. The composite reinforced with 15 vol.% showed better cellular response compared to its 30 vol.% counterpart (see Figure 3b, where SEM and fluorescence of initial cell morphology is showed). This was explained as due to an increase in the pH of the medium at a high percentage of magnesium. In the degradation analysis of composites, evaluated in simulated body fluid (SBF), pH reached values of 9.5 for PLA-30 Mg. In a novel way, in the same study, authors analyzed the antibacterial capacity of magnesium ions under near infrared (NIR) emission based on previously observed photothermal studies [93,94]. They evaluated the antibacterial capacity of PLA/Mg composites under the Escherichia Coli (gram-negative) and Staphylococcus Aureus (gram-positive) bacterial study by applying hyperthermia on the systems. During two minutes of emission at a wavelength of 808 nm (400 mW), the authors observed a hyperthermia effect (Figure 4). This effect, applied on bacterial culture plates, showed antibacterial capacity (see again Figure 4c–f) with both percentages of reinforcement, thus confirming the enhancement of the antibacterial character under NIR emission. The authors suggested that by applying photothermal agents to bone implants, NIR irradiation can be used to generate localized heat (hyperthermia phenomenon), with sufficient capacity to kill nearby pathogenic bacterial activity [94]. In reference to mechanical studies, the samples doped with 15 vol.% reinforcement showed better mechanical performances with respect to the system doped with 30 vol.% reinforcement. The authors explained that this behavior is due to a low affinity between reinforcement and polymeric matrix which affects in mechanical terms the performance of the cylinders [95].

#### 4.1.2. PLA Reinforced with Mg Wires

The improvement of PLA mechanical properties by incorporating Mg with the particle form is limited, and bone reparation required adequate and suitable mechanical properties. Compressive strength can be enhanced, but tensile strength could decrease [86]. The strengthening effect under compressive testing competes with the effect of particles on the thermal degradation of the polymer and on the interference in the crystallization of the matrix. Strategies to effectively improve the tensile and/or flexural strength of PLA/Mg composites suggest the incorporation of Mg alloys in the form of braided or unidirectional wires (see Figure 5). The most common alloys that have been used as wire reinforcement for PLA are AZ31 and MgZn.

In this sense, Li, X. et al. [96] reinforced PLA (density of 1.24 g/cm^3^) with 2D braided wires of AZ31 alloy with a diameter of 0.3 mm (Figure 5b). The effect of the braiding angle and wire content on the mechanical properties of the composite was evaluated. They studied four different braiding angles (15, 30, 45 and 60°) at 10 vol.% and four different wire contents (5, 7, 10 and 12 vol.%) at a braiding angle of 30°. The fabrication process of the composite consisted of lamina stacking. Laminas were formed by pouring a PLA solution in chloroform onto the braided wires. PLA/AZ31 composite laminas and PLA laminas were alternative stacked and blended by hot pressing. They found that the tensile and bending strength of the composite decreased proportionally with the braiding angle. The shear strength and impact strength showed a maximum value for the composites reinforced with wires braided at 45°. Regarding the effect of the volume fraction of wires, tensile, bending, shear strength and impact strength increased with AZ31 content. The manuscript also studied the effect of the modification of wires surface by micro-arc oxidation (MAO). The modification allowed a stronger filler–matrix interface that induced a positive strengthening effect on tensile, bending, shear and impact strength. For instance, the increment in tensile strength after MAO treatment was close to 20% for a volume fraction of 10%. In the same way, the effect of wire content on the mechanical properties of the composite has been studied by Ali, W et al. [99]. They fabricated composites using unidirectional AZ31 wires and PLA (Grade 3251D, 90,000–120,000 g/mol). Composites with different wire volume fractions (0 as the control, 20, 30, 40 and 50%) were also prepared by lamina stacking. The mechanical properties were studied by tensile and flexural tests. Tensile and flexural strength increased with Mg wire content. Force displacement graphs of tensile tests together with the fracture morphologies of tensile and flexural tests are shown in Figure 6a.

The study of the effect of different treatments on the surface of wires on tensile and bending strength in PLA/Mg composites has been also performed by Cai, H. et al. [100]. In this case, they prepared composites using Mg2Zn wires with a diameter of 0.3 mm and PLA (Grade 3251D and 78,000 g/mol). They treated the surface of Mg2Zn wires with MAO, dopamine (PDA), hydrofluoric acid (HF) and a silane coupling agent (KH550). Mg2Zn wires/PLA composites were prepared again by lamina stacking with 15 vol.% wire content for tensile tests. Load vs. displacement curves are shown in Figure 6b. The highest tensile strength was achieved with Mg2Zn wires treated with MAO and HF (88 ± 3 and 85 ± 3 MPa, respectively). Bending strength was evaluated in composites with different volume fractions of wires (0 as the control, 5, 10, 15 and 20 vol.%) in untreated conditions and treated with MAO, PDA, HF and KH550 (results are shown in Figure 6c). The highest bending strength was achieved with the highest volume content (20 vol.%) and MAO treatment. Authors explained that the interfacial bonding strength depends on the interfacial constitute and microstructure and evaluated the interaction energy. MAO treatment generates MgO on wires surface and HF treatment generates MgF_2_. The better mechanical performance with MAO and HF treatments is because the interaction energy (EI) in the MgO-PLA interface is approximately 2500 times higher than that in Mg-PLA, and in the MgF_2_-PLA interface, it is close to 50-times higher than that in Mg-PLA.

As mentioned earlier, adequate tensile and flexural strengths are important for bone repair applications that imply load bearing, but more relevant for this type of applications is the evolution of the mechanical properties of the material when it is subjected to a physiological environment. Once the optimal reinforcement is achieved, it is important to study how the evolution of the mechanical properties of the composite is while it degrades. Table 3 summarizes the studies performed on the degradation of PLA/Mg composites reinforced with wires and gives information regarding the evolution of bending strength with immersion time in a physiological medium.

Ali, W. et al. [99] compared the degradation of the mechanical properties of a PLA composite reinforced with 50 vol.% of AZ31 wires vs. pure PLA. They performed the degradation tests in PBS at 37 and 50 °C and studied the tensile strength and flexural strength at certain immersion times (0,1, 4, 7, 14, 21, 28 and 35 days). They found that the elastic and flexural moduli of the composite were stable during the first four weeks, which is an important property for load-bearing applications. They also found that the degradation of the composite caused an increment in pH, which at 37 °C stayed below 8 for the first 21 days and increased up to 9 at the end of the experiment (35 days). At 50 °C, pH remained below 8 during the first 2 days and reached the value of 11 after 14 days. The authors stated the importance of controlling the degradation rate of the composite by different surface treatments. Therefore, further research works from Ali, W. et al. focused on the effect of surface treatments of wires on the degradation behavior of PLA/Mg wire composites.

The effect of fluoride coating on the degradation behavior of a PLA/AZ31 composite reinforced with 50 vol.% of wires has been studied [97]. The mechanical performance of a composite without the surface treatment and another coated with fluoride has been compared. Degradation tests were performed at 37 °C in PBS solution for 8 weeks (56 days). Composites were tested under tensile and flexural loading. Surprisingly and in contrast with the studies of Cai, H. et al. [100], fluoride coating did not improve the initial mechanical properties of PLA reinforced with untreated wires. However, the fluoride coating was effective in controlling the decrement of mechanical properties during the first 4 weeks. Tensile strength slightly dropped during the first day (163.7 MPa to 156.5 MPa), but then it remained at stable values until day 28 and further decreased till day 56. The composite reinforced with untreated AZ31 wires suffered a drastic decrement in tensile strength and lost its mechanical integrity after 35 days immersed in PBS. Fluoride treatment was also effective in maintaining the elastic modulus of the composite for 4 weeks and in slowing down the loss of flexural strength with time.

Modifying the surface of Mg wires is not the only strategy that has been studied to slow down the loss of mechanical properties. In the same way, Cai, H. et al. [102] studied the effect of modifying the polymeric matrix by hot drawing on the in vitro degradation behavior of composites. They prepared samples based on PLA (Grade 3251D, 78,000 g/mol) and reinforced with Mg2Zn in a content of 0 as the control, 5 and 10 vol.%. In this case, wires were treated with MAO. Composites were prepared by lamina stacking and hot pressing and subjected to subsequent hot drawing to induce self-reinforcement in the polymeric matrix by the orientation of PLA chains. Bending strength was measured at different immersion times. Samples were immersed in Hank´s solution at 37 °C for 2, 4, 6, 8 and 10 weeks. The effect of the wire content and hot drawing on the degradation of composites was analyzed as well as the mutual influence between Mg wires and PLA in the degradation of the composite. Hot drawing increased the crystalline fraction of the PLA matrix, and therefore, the initial bending strength of composites increased with hot drawing and with the wire content (see again Table 3). Regarding the degradation behavior of the composite, the polymeric matrix with higher orientation and crystallinity hindered the diffusion of water from the surface to the inner part of the composite, slowing down the degradation rate of the composites and decreasing the corrosion rate of Mg2Zn wires embedded within the matrix. On the other side, higher wire content accelerated the degradation rate of PLA as Mg presence catalyzed the hydrolysis of the polymer. As a result, the bending strength of composites decreased slowly during the first 6 weeks and afterwards it decreased quickly.

The effect of dynamic compressive loading on the degradation behavior of PLA/Mg wire composites has been studied by Li, X. et al. [98,101]. They prepared composites based on PLA (1.24 g/cm^3^) and reinforced with AZ31 wires. In these studies, they evaluated again the effect of dynamic compressive loading on the degradation behavior of pure PLA and a composite reinforced with 10 vol.% of wires [101] and of a composite reinforced with 20 vol.% of wires [98]. In both cases, wires were treated with MAO to improve the interface adhesion. Materials were immersed in Kirkland´s biocorrosion media (KBM) at 37 °C, and the bending strength was evaluated. They concluded that dynamic compressive loading increases the degradation rate of PLA and composites. The bending strength decreases with time, with the increment in frequency and with the increasing loading magnitudes [98,101]. Bending strength retention depends on the conditions of the dynamic loading. For instance, less than 20% of the bending strength of PLA reinforced with 20 vol.% of AZ31 wires remained after 8 weeks at 1 MPa and 1 Hz, while 50% of its bending strength is retained at 0.2 MPa and 0.5 Hz [98] (Figure 6d). Pure PLA retained 50% of its bending strength after 21 days at 0.9 MPa and 2.5 Hz, while PLA reinforced with 10 vol.% of AZ31 wires retained 60% [101]. The authors state that this type of composite could provide enough mechanical stability for 8 weeks for a successful bone fracture healing.

In a complementary manner, Sun, S. et al. [103] developed PLA/Mg porous composites based on PLA (100,000 g/mol) reinforced with AZ31 wires. The authors obtained the porous matrix by a fabrication process that consisted of pouring PLA solution into a mold with ice particles and subsequent lyophilization. Mg wires were assembled into channels placed strategically in the porous PLA matrix. Before placing the wires, they were immersed in SBF to induce the deposition of a biomimetic coating containing calcium and phosphorous. The authors studied the proliferation of adipose-derived stem cells (ADSCs) in scaffolds with treated AZ31 wires (PLA-MC), scaffolds with untreated wires (PLA-Mg) and PLA scaffolds. They found that PLA-MC scaffolds accelerated cell adhesion and proliferation capacity with respect to the control system; in addition, the system showed synergy with the Ca-P coating, slowing down the degradation process and allowing sustained release of magnesium ions over time.

#### 4.1.3. 3D Printing of PLA/Mg Particle Composites

Additive manufacturing (AM) technologies have been a breakthrough since their introduction due to their potential advantages, such as the customization of designs and the ability to create complex structures in one piece and without assemblies, reasons that favor the fabrication of complex geometry as interconnected scaffolds, ideal in TERM studies [104,105,106]. Methods of obtaining PLA/Mg composites by 3D printing can also be found in the literature. In fact, PLA is one of the most used and favorable polymers for 3D printing based on FFF technology due, among other things, to its low melting point [107]. For example, Bakhshi, R. et al. [108] describes the fabrication of scaffolds by FFF technology of PLA matrix and reinforced with magnesium (≤100 µm, irregular flake) for percentages of 0 as the control, 2, 4, 6, 8 and 10 wt.%. The mixing process was developed by dissolving the polymer together with the reinforcement in dichloromethane (DCM) with stirring for 5 min. Then, the mixture was heated in an oven for 12 h at 80 °C and then passed through an extruder to promote homogeneity (rotation at 30 rpm and 190 °C). The filament obtained was printed at a speed of 30 mm/s with 210 and 55 °C (extrusion and bed temperatures, respectively) generating the structures shown in Figure 7a (porosity close to 47% and 800 µm pore size). By means of FESEM images, it was possible to observe an increase in terms of filament roughness with the percentage of incorporated magnesium. In addition, porosity close to the reinforcement particles was observed in the samples with higher metallic percentage (see also Figure 7a). The authors discussed the pores associated with problems derived due to the impediment effect that metallic particles cause in the PLA flow during the printing process. On the other hand, DSC analysis showed a slight increase in Tg values with the presence of magnesium, i.e., it is known that the reinforcements usually reduce the mobility of the associated polymeric chains, having the direct consequence of an increase in the transition values [109]. A degradation study confirmed that the metal presence accelerates the process as other authors had confirmed [91,110] based on an increase in the corrosion products and in hydrophilicity. In vitro studies were performed with the human fibroblast cell line (L929). In this test (Figure 7d), the presence of magnesium improved the adhesion and cell proliferation of the control system, with a maximum percentage of 6 wt.%, where an excessive presence of magnesium ions may have a negative effect on the fibroblast culture (observed from 8 wt.%).

Similarly, Ali, F. et al. [111] complemented the study of Bakhshi, R. et al. [109] of composite printing under FFF technology but in this case with magnesium alloy reinforcement (AZ61). The PLA matrix used (3–5 mm pellet of approx. 68,000 g/mol) was diluted in chloroform at 20 g/L ratio and 400 rpm for 4 h at room temperature. The alloy was added to this mixture as reinforcement in different percentages: 0 as the control sample, 5, 10 and 15 wt.% (diameters <100 µm). Then, after drying at room temperature for 24 h, the authors passed the mixture through the extruder (175 °C), obtaining filaments of 2.75 mm diameter. From SEM images, the fabricated composites showed good distribution and no observable defects. On the other hand, as in previous studies, the presence of the metallic reinforcement increases the Tg value as it is an impediment in terms of intermolecular mobility. In terms of degradation, the samples were immersed for 2–4 weeks in PBS, and degradation was observed in proportion to the percentage of reinforcement incorporated and associated with an opportune increase in the pH of the medium (values of 8.5 with only 5 wt.% of reinforcement). The in vitro qualitative study developed on MC7s epithelial line showed improvements in terms of cell adhesion under magnesium alloy reinforcement. Also based on 3D manufacturing technology, it is important to highlight the article developed by Zeynivandnejad, M. et al. [112], in which the importance of the proposed printing direction (0, ±45, 45 and 90°) on the final properties of the material (up to 1 wt.% reinforcement) was analyzed. By DSC analysis, again a slight increase in the Tg value with the presence of magnesium was demonstrated, while in mechanical terms, in the 0° direction, i.e., parallel to the test direction, the samples showed higher strength. As a complementary study, degradation study developed in SBF showed that the presence of magnesium favors its degradability, being more pronounced in directions where the melting surface has more contact with the medium, i.e., 90° printing condition.

Finally, printing processes that respond to the limitations of traditional FFF technology have also been addressed. Schmidt, F. et al. [113] presented a preliminary study on powder bed fusion-laser based (PBF-LB) technology. This technology, unlike FFF, allows the printing of geometries with better resolution and without the requirement of supports [114]. Thus, PBF-LB technology is based on laser sintering of polymer powders, whereby the final characteristics are dependent on the type of laser applied and the powder particle of origin [115]. In the manuscript, 3–5 mm PLA pellets were used as the matrix and reinforced with partially oxidized magnesium powder (<40 µm) with different percentages (0 as the control, 10, 25 and 40 wt.%). The mixture of both elements was developed by cryogenic milling at 8500 rpm in the presence of liquid nitrogen. In reference to the printing process, the printing was developed with a CO_2_ laser (14 W), laser speed of 3500 mm/s and temperature of 115 °C. The resulting material, under SEM analysis, showed how the reinforcement was correctly dispersed and incorporated in the polymeric matrix, although it also showed a high percentage of porosity that the co-authors associated to a low packing density during the printing process that argued the high brittleness of the samples obtained with the presence of Mg.

In a complementary manner and in order to facilitate the information provided to the reader, a brief summary of PLA with Mg particle reinforcement is incorporated in Table 4. Similarly, and also for the reader’s convenience, Table 5 shows a summary of the biological preclinical evaluation of the bioabsorbable composites reported with applications in bone tissue engineering.

### 4.2. Bioabsorbable Hybrid Materials Based on PCL Reinforced with Mg (PCL/Mg)

The study and fabrication of PCL-based systems reinforced with Mg to form bioabsorbable composite materials is, as PLA-based composite, a novel technology that emerged in the 2010s. The limited properties of PCL as a bioabsorbable element such as its mechanical capacity and/or low degradation rate encourage the incorporation of reinforcing metallic elements such as Mg to improve its performance in the biomedical field. In this section, the works found in the literature will be organized in different subdivisions where PCL will be reinforced with particles. In the first point, the composites are fabricated by traditional methods, and in the second point, the fabrication method is based on 3D printing. Finally, a final section is included about novel PCL copolymers with Mg reinforcement.

#### 4.2.1. PCL Reinforced with Mg Particles

The effect of Mg particle size and surface modification on the release of Mg ions, compressive mechanical properties and cytocompatibility in in vitro and in vivo studies of PCL/Mg composites has been addressed by Wong, H. M. et al. [116] (see Figure 8). They developed their study systems based on PCL (approx. 80,000 g/mol) and reinforced it with Mg microparticles with different sizes (45 and 150 µm). The surface modification of the reinforcement was developed with the aim to improve the bonding generated between the metal and the polymeric matrix. The particles were coated with silane-based common coupling agents such as 3-(Trimethoxy silyl) propyl methacrylate (TMSPM) [117]. Already coated, particles were mixed in a Mg/PCL ratio of 0.1:1 (60 °C) with the polymeric matrix. The presence of silane and the variability in the size of the reinforcement did not show significant differences in the release of magnesium ions (see Figure 8a). In terms of stiffness under compression testing, the silane-reinforced samples also showed better mechanical performance, around 20% higher than their counterparts (Figure 8b), thus demonstrating their efficiency in the application of coupling agents between both systems and no influences on size reinforcement. To analyze the cell viability of these materials, pre-myoblast cell line MC3T3−E1 was evaluated and showed in Figure 8c. All the materials studied, including the control, showed cytocompatibility, with a 40% improvement when the samples were reinforced with magnesium. In relation to the presence of magnesium ions and capacity to promote bone tissue synthesis, the protein marker alkaline phosphatase (ALP) was evaluated, showing significant differences with respect to the control evaluation (PCL pure and PMMA) in the assays analyzed at 14 days. The highest ALP activity was shown by the composite reinforced with particles of 45 µm and modified with TMSPM. In the case of in vivo evaluation, the researchers based their study on the use of 2-month-old female Sprague–Dawley rats (SD rats), which were implanted with the new material in the lateral epicondyle of the animal. Figure 8d shows the influence of Mg particle incorporation and their surface modification on bone formation (red arrow in the image) quantified by micro-CT. Greater osteoconduction was shown in magnesium-reinforced samples compared to the proposed control systems without any associated inflammatory response (Figure 8e).

As a continuation of this research, Wong, H. M. et al. [118] fabricated PCL/Mg systems with porous and interconnected morphology to generate scaffolds for bone tissue applications. The process of obtaining the scaffold was developed using the salt leaching technique, in which, in brief, 1 g of PCL (approx. 80,000 g/mol) was diluted in 10 mL of dichloromethane (DCM). Then, with 10 mL of absolute ethanol (EtOH), the previously used solvent was removed, and magnesium particles (45 and 150 µm) were added up to 20 wt.%. NaCl (7 mg) was incorporated to the mixture, and this was poured into a mold. Finally, the solvent evaporated overnight, obtaining the final material, which after washing in NaOH solution favors the leaching of NaCl, obtaining a highly porous structure (approximately 74% by micro-CT analysis). The presence of magnesium particles provided a higher elastic modulus and cell viability with the MC3T3−E1 cell line compared to the control sample. Mg microparticles were shown to enhance cell adhesion and proliferation, thus improving the overall cellular activity of PCL/Mg scaffolds. For the in vivo study, they used the same mouse model as in the previous work for comparative purposes. The analysis developed for 3 months showed degradation and synthesis of bone apatite in all cases, which was higher when magnesium content was present. Thus, the high importance of magnesium ions in terms of bone tissue synthesis was recognized.

Based on the production process under the electrospinning technique, PCL nanofiber-based mats were obtained by Salaris, V. et al. [119] and also reinforced with functionalized magnesium: Mg(OH)_2_ and MgO (10 and 20 nm) in percentages of 0, 0.5, 1, 5, 10 and 20 wt.%. Prior to the electrospinning process, the mixture was prepared from a solution of PCL in chloroform:DMF (4:1) solvent and then sonicated in the presence of the nano-reinforcement. The manufacturing process conditions were positive and negative voltages set at −10 and 10 kV with flow rates of the polymer solution and the solvent solution fixed at 1.0 and 0.5 mL/h, respectively. The choice of magnesium hydroxide was based on its osteoinductive capacity and greater stability than Mg particles [120]. As opposed to metallic reinforcement, Mg(OH)_2_ in contact with the physiological medium does not form H_2_ bubbles that can be harmful in areas close to the implant [121,122]. For the second, MgO is largely used as a reinforcing agent due to the high surface reactivity, chemical and thermal stability in conjunction with antimicrobial and antitumor performance [123,124]. From SEM images, it could be observed that magnesium hydroxide samples showed better dispersion compared to MgO samples; on the other hand, the diameter of PCL fiber obtained increased with increasing reinforcement content. Moreover, under DSC analysis, no differences were observed in terms of the melting temperature value, and a reduction in the crystallinity value associated with the matrix polymer was observed. Finally, the presence of nano-reinforcements improved the mechanical performance compared to that of the control system.

#### 4.2.2. 3D Printing of PCL/Mg Composites

As was the case for using PLA as a matrix, different studies based on AM can be found in the literature. The effect of Mg content on the surface roughness, cell adhesion and proliferation in in vitro and in vivo studies in PCL/Mg composites has been addressed by Zhao, S. et al. [125]. They used FFF technology to manufacture polymeric scaffolds based on PCL (Mn approximately 68,000 g/mol), reinforced with Mg particles of 45 µm size. In order to obtain a uniform mixture, the precursors were heated at 100 °C in a biaxial roller mixer. Scaffolds with three percentages of reinforcement (5, 10 and 15 wt.%) together with a comparative control were evaluated and printed under the following conditions: temperature 110 °C, gas pressure 0.6 MPa and printing speed of 6–8 mm/s. All the proposed scaffolds presented the desired printability (see Figure 9); however, according to the images analyzed by SEM, the presence of magnesium in the PCL causes surface modifications that affect the final roughness of the part (Figure 9a). The authors attributed the heterogeneities on the surface to the dissimilar melting points and cooling capacities of PCL and Mg. In fact, the contact angle studies (Figure 9b) showed that the presence of magnesium in the filament favors the surface hydrophilicity of the system with more than favorable consequences in biological studies, where it is recognized that hydrophobic systems tend to hinder elementary processes such as adhesion and cell proliferation in TERM studies [126,127]. On the other hand, the release of magnesium ions showed that the samples with a higher presence of magnesium released a greater number of ions with a consequent increase in pH medium, reaching in samples with 15 wt.% alkaline values of pH 8.3 after three days of study. This result was supported by the in vitro assay developed from the rat bone marrow mesenchymal stem cells (rBMSCs) cell line, where samples with 10 wt.% of reinforcement showed the highest capacity for cell adhesion and proliferation. In addition, in terms of differentiation towards bone lineage, the quantification from ALP marker also reflected that the samples reinforced with 10 wt.% allowed greater development with respect to 0, 5 and 15 wt.%. The authors argued that the worse performance at 15 wt.% was due to the high pH generated by an excess of magnesium ions released, which counteracts the osteoinductive capacity of the reinforcement. Finally, the in vivo assay was evaluated with SD rat defective skull models. The results obtained based on micro-CT and X-ray measurements showed a higher bone regeneration capacity of the 10 wt.% magnesium-reinforced system compared to the control system (Figure 9c). The associated null toxicity was demonstrated by histological studies in tissues such as liver, kidney or heart.

The effect of Mg content on the mechanical properties, in vitro cytocompatibility studies and in vivo bone regeneration capacity has been also studied by Dong, Q. et al. [128]. They fabricated scaffolds using FFF technology. They developed their material based on PCL (80,000 g/mol) at 200 °C blended with magnesium particles (28.6 µm) in different percentages (0, 1, 3, 5, 7 and 9 wt.%) for 60 min. The printing conditions were a temperature of 160 °C and printing velocity of 1.5 mm/s. The scaffolds obtained showed a 66% porosity with a porosity size of 480 ± 25 µm. The analyses referred to DSC studies showing a reduction in the degree of crystallinity of PCL with the increment of Mg content especially from 3 wt.%. Also, an increase in hydrophilicity was observed with Mg contents higher than 3%. In terms of compression studies, scaffolds with 3, 5 and 7 wt.% of Mg exhibited higher compressive resistance than the control or with low content reinforcement. The appropriate Mg incorporation has a reinforcing effect on mechanical properties, but excessive amounts of Mg micro-particles weaken the mechanical performance. Cytocompatibility studies were based on the rBMSC cell line. A progressive improvement in cell adhesion and proliferation capacity was observed up to 5%. Higher content of Mg reinforcement implied an excessive release of magnesium ions, which reduced the effectiveness of the scaffold. Under cell quantification with ALP, it was observed that percentages of 3% generated greater apatite synthesis capacity, showing significant improvements with respect to the control system. Finally, in vivo studies, based on white rabbits of about 3–5 months, were evaluated in localization on the medial tibial tubercle and confirmed that the 3% reinforcement systems had greater bone regeneration capacity than the control systems. 

Abdal-hay, A. et al. [18] reduced the size of the reinforcement (below 50 nm) and the nano-reinforcement was functionalized with hydroxyl groups, Mg(OH)_2_, which were named MH. They produced PCL/MH scaffolds using FFF technology. For the matrix, PCL (45,000 g/mol) was selected, and the mixture was made from a solution in chloroform (10% *w*/*v*) with MH (5 and 20 wt.%) in an ultrasonic bath for 30 min plus casting of the mixture for 72 h to eliminate the organic solvent. The mixture was printed under conditions of 95 °C, air pressure of 550 KPa and printing speed of 4 mm/s. After the printing process, the scaffolds were immersed in 70% wt./vol. EtOH and kept under vacuum for 15 min to remove the residual solvent and/or trapped air bubbles. A posteriori, the authors immersed the samples in 3 M sodium hydroxide (NaOH) at 37 °C plus dried under vacuum for 15 min. Finally, the samples were washed with ultrapure water to remove unwanted debris. The inclusion of NPs had hardly any influence in terms of crystallinity, but it did have an influence on the mechanical properties as Salaris, V. et al. [119] cited, where the samples with 5 wt.% reinforcement showed better strength in the tensile and compression tests evaluated with respect to the control sample and the one reinforced with 20 wt.%. The authors explained this result based on the increment in the percolation limit with higher particle contents. In terms of in vitro evaluation, isolated human osteoblast (hOB) cell line was evaluated, and the nanoparticles favored greater capacity of adhesion and cell proliferation with respect to the control system. In addition, in relation to differentiation processes, protein markers measured with ALP also showed a greater capacity of cell differentiation towards bone lineage with nano systems reinforced with MH.

Calcium-based biomaterials, due to their good overall biocompatibility, osteoconductivity and similarity with respect to the inorganic component of natural bone, have been widely used in biomedical applications [129,130]. With the intention of forming synergy with magnesium particle reinforcement towards regeneration of bone lesions, Tsai, K.Y. et al. [131] developed a reinforcement mixture between calcium silicate and magnesium (CS-Mg). They studied the effect of different contents of CS-Mg powder in PCL (0 as a control, 10, 20 and 30 wt.%) on the hydrophilic character of the system, degradation rate and in vitro cytocompatibility. In this case, the authors proposed a printing process for scaffolds using laser sintering technology (LS). This technology represents a versatile 3D printing mode based on the coalescence of powders from a sintering process that allows obtaining parts with resolutions higher than FFF [132]. In terms of hydrophobicity, the reinforced samples generated an increase in the hydrophilic character of the systems together with a higher degradability after 12 weeks of study due to the higher degradable speed of the reinforcement with respect to the polymeric matrix. Cytocompatibility studies were developed with the hMSC line. The samples showed a greater osteoinductive character with the higher content of reinforcement after analysis with markers measurements such as Alizarin red and osteocalcein. The authors described, in this case, the synergy formed with the corresponding release of silicate ions, which has an essential role in bone-forming metabolic processes [133,134], with the response of the influence of magnesium ions in regeneration of tissues of a bony nature. 

As it was mentioned with PLA/Mg and in order to facilitate reading, Table 6 summarizes the studies described in Section 4.2.1 and Section 4.2.2, indicating the materials used, the amount of reinforcement and processing method. Table 7 shows the preclinical work developed on this bioabsorbable composite for bone tissue engineering applications.

#### 4.2.3. Mg-Reinforced PCL-Based Copolymers

Finally, it is worth mentioning other studies in which the polymeric matrix is based on PCL copolymers. In the case of Shen, J. et al. [135], they developed a 3D printed material with a polymeric matrix based on polycaprolactone-co-poly(ethyleneglycol)-co-polycaprolactone (PCL-PEG-PCL) copolymers with a surface modified by the presence of magnesium oxide (MgO) nanoparticles. This reinforcement improved the wettability of the control system and its resistance capacity in compression tests. In terms of in vitro analysis, under the MC3T3−E1 cell line, previous works described the presence of magnesium that also represented improvements in cell adhesion, proliferation and differentiation towards the bone lineage with respect to the control systems. Animal studies were performed in thirty female Sprague–Dawley, which were 12-weeks-old. The presence of MgO stimulated the regeneration process, but the excess of metal reinforcement (10:1 wt. ratio between copolymer and MgO) limited the efficiency of the process. In another study, PCL/Mg based systems have also been used as reinforcements for collagen polymeric membranes [136], characterized by a high degradation rate and a weak structure. In this study, Wang F. et al. [136] fabricated a system based on photopolymerizable interpenetrated (IPN)-type structures that improve the response in terms of mechanical properties and in the in vivo studies developed, where the presence and reinforcement of Mg at 20 wt.% favored the generation and processes of cell differentiation towards bone tissue.

## 5. Concluding Remarks and Future Perspectives

After more than a decade of research on PLA/Mg and PCL/Mg composites, a large amount of empirical data has been collected that allows the validation or modification of the proposed hypotheses. Related with the preceded points, the idea that Mg could provide mechanical reinforcement to the polymeric matrix is valid for some combinations of matrix and types of reinforcement. The enhancement of mechanical properties depends on the balance between the strengthening effect of Mg particles and their influence on the thermal degradation and crystallization of the polymeric matrix. Particles can improve the behavior of the matrix under compression but deteriorate the mechanical performance under tensile or flexural stresses. The amount of Mg particles that can be incorporated into the polymeric matrix depends on the manufacturing technology and the matrix–filler interaction. Metal–polymer bonding can be improved by surface treatments on the reinforcement or by adding coupling agents in order to enhance the stiffness of the material under compression testing. Nonetheless, excessive amounts of Mg microparticles weaken the mechanical performance, and a strengthening effect is obtained only with an appropriate amount of reinforcement.

The stability of the compressive strength with time of particulate composites depends on the shape of the reinforcement. Irregularly shaped particles degrade faster than spherical ones. Composites reinforced with irregularly shaped particles lose their mechanical integrity in a week, while composites reinforced with spherical ones retain 60% of their compressive strength after 4 weeks. Relevant for load bearing applications is the capacity of the material to retain its mechanical properties during the first 4 weeks in physiological conditions. Therefore, for particulate composites to be suitable for load-bearing applications, further studies are needed to control the loss of their compressive strength.

On the other hand, to effectively improve the tensile strength and bending strength of composites, the reinforcement with Mg wires has been proposed. This strategy has been considered mainly with PLA; there are no studies on PCL reinforced with Mg wires. The surface modification of wires allows a stronger filler–matrix interface improving the mechanical performance of the material. MAO and HF treatments are the most effective. Tensile and bending strength can be tailored by changing the Mg wire content and surface treatment. PLA/Mg composites reinforced with wires have the capacity to retain their elastic and flexural moduli during the first 4 weeks in physiological conditions. This implies that this type of composite could provide enough mechanical stability for successful bone fracture healing.

Concerning composite degradation, it has been confirmed that the polymeric matrix plays an important role in controlling the degradation rate in physiological environments of Mg reinforcement. Regarding the effect of the crystalline degree on the degradation rate of composites, contradictory results have been obtained between composites reinforced with particles and composites reinforced with wires. While a high crystalline matrix accelerates the degradation rate of particulate composites, a polymeric matrix with higher orientation and crystallinity slows down the degradation rate of Mg wires in composites reinforced with them. Then, further research on the role of the crystallinity of the polymeric matrix in polymer/Mg composites is needed to elucidate the reasons behind these contradictory results. It is known that high molecular weight polymers degrade at a slower rate than low molecular weight polymers. Therefore, it is expected that the molecular weight of the polymeric matrix (PLA or PCL) influences the bioabsorption rate of composites reinforced with Mg. However, there is a lack of information on this topic. Studies included in this review mention the molecular weight of the polymer used but do not analyze its role on in vitro or in vivo degradation rate. Research on the role of the molecular weight of PLA and PCLon polymer/Mg composites bioabsorption rate is needed to fully comprehend the degradation behavior. On the other side, it has been demonstrated that Mg content catalyzes the hydrolysis of the polymer, accelerating its degradation. It is therefore important to balance the reinforcement effect of Mg with its influence on the increment of the degradation rate in the composite, in order to find applications in load-bearing resorbable implants and tissue engineering.

The hypothesis concerning the enhancement of the bioactivity of bone stimulation by the incorporation of Mg reinforcement has also been validated in the in vitro studies described in both matrices (PLA and PCL). Mg ions released in physiological environments promote bioactivity and osteoconductivity. It has also been observed that cell adhesion and proliferation capacity improve with respect to pure polymers. However, an excessive presence of magnesium ions in cell culture media may have a negative effect on cell proliferation and bone stimulation. This has been explained by different authors by the increment in pH medium with high ion Mg content. The Mg content stated as “excessive” reduces the effectiveness of the cell proliferation process and depends on the nature of the polymeric matrix, the type of sample (bulk or scaffold) and the processing technology.

Regarding in vivo evaluation, it has been confirmed that the presence of Mg in a polymer stimulates the bone formation process, accelerates the synthesis of bone apatite, reduces the inflammatory response and enhances osteoconduction in comparison to systems without reinforcement. However, these results have only been validated for PCL/Mg scaffolds. There is a lack of in vivo evaluation for PLA/Mg systems. Further research on the in vivo performance of PLA/Mg composite systems is needed to validate this material for future clinical applications in bone therapy. In the case of load-bearing applications, polymeric matrix reinforced with Mg wires look more appropriate (although to this date, only studies based on PLA/Mg were found), while particulate composites could be useful for tissue engineering and low load-bearing applications.

On the other hand, studies based on functionalized reinforced systems or mixed with other elements such as calcium silicate have shown synergy in terms of osteosynthesis, so that the combination of reinforcements could also lead to improvements with respect to control patterns. Finally, the antibacterial capacity shown in this work provides the magnesium reinforcement with, in addition to improvements in terms of mechanical and osteoinductive properties, the ability to produce bacterial death from hyperthermia. As described earlier, due to NIR irradiation, the metal raises its temperature, allowing the death of surface biofilm, and thus is an ideal complement in bio applications.

Although PLA/Mg and PCL/Mg systems are promising bioabsorbable composite materials for applications in bone regeneration, the final validation of these systems in the medical field needs to be supported on further research. As the technology becomes more accessible and optimizable, the associated difficulties will allow greater adoption at the clinical level based on the collaboration of the different research groups involved.

## Figures and Tables

**Figure 1 polymers-15-04667-f001:**
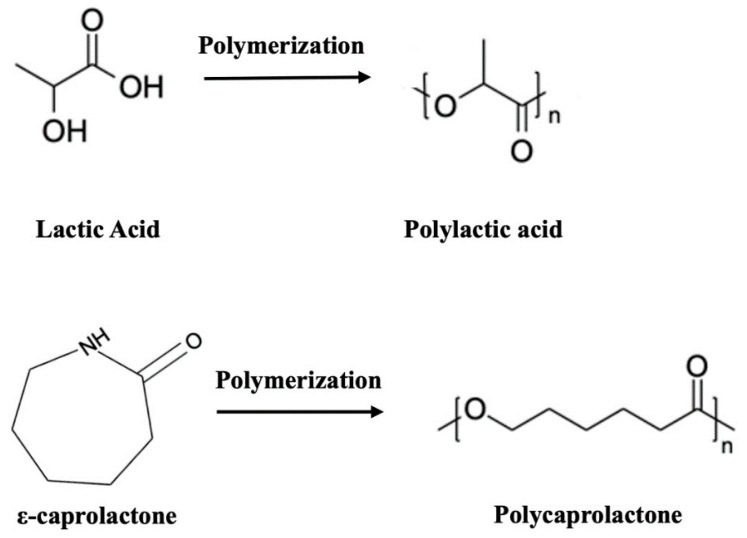
Structures of the bioabsorbable polymers involved: PLA and PCL with their associated monomers.

**Figure 2 polymers-15-04667-f002:**
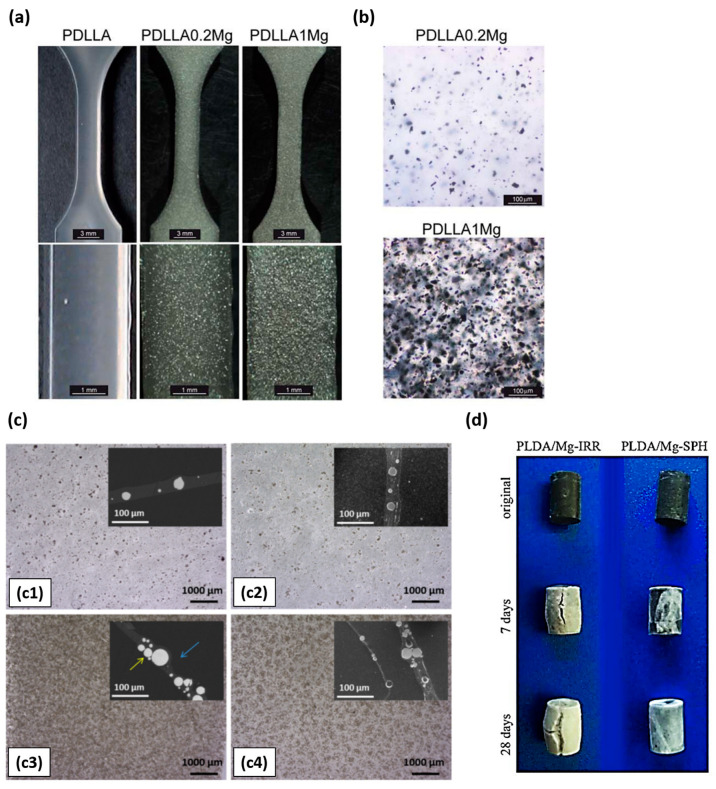
(**a**) Macrographs of PDLLA and PDLLA/Mg injection-molded samples (scale bar 1 mm); image adapted from [86]. (**b**) Visualization of Mg particles in PDLLA/Mg composites using polarized light microscope (scale bar 100 μm); adapted from [86]. (**c**) Surface micrographs of PLA/Mg films prepared by colloidal processing reinforced with 10 wt.% Mg particles (**c1**,**c2**) and 50 wt.% Mg particles (**c3**, where blue and yellow arrows mark the influence of load on surface and **c4**) modified with PEI (**c1**,**c3**) and CTAB (**c2**,**c3**); adapted from [88]. (**d**) Photographs of PDLLA/Mg composites processed by hot extrusion and compression molding reinforced with 10 wt.% irregularly shaped Mg particles (Mg-IRR) and spherical Mg particles (Mg-SPH) as processed (original) and after 7 and 28 days immersed in PBS; adapted from [89].

**Figure 3 polymers-15-04667-f003:**
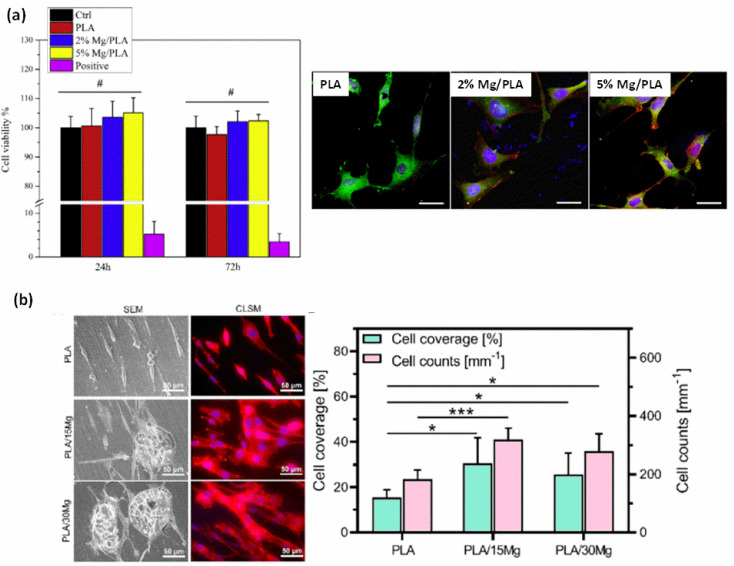
(**a**) MC3T3−E1 cell viability after 24 and 72 h incubation in different extracts (# *p* > 0.05) and representative confocal microscopic images of cell staining of actin filament (red), vinculin (green) and nuclei (blue); white scale bar = 50 µm and image adapted from [91]. (**b**) Initial cell attachment (MC3T3−E1) images of samples: SEM and fluorescence of initial cell morphology; cell per unit area after 24 h of culturing pre-osteoblast cells; white scale bar = 50 µm and statistical differences were evaluated by (* *p* < 0.05 and *** *p* < 0.005). Adapted from [92]. This article was published in Biomater. Adv., 152 (May), 213,523, Lee, H et al. Antibacterial PLA/Mg Composite with Enhanced Mechanical and Biological Performance for Biodegradable Orthopedic Implants. Copyright, Elsevier, 2023.

**Figure 4 polymers-15-04667-f004:**
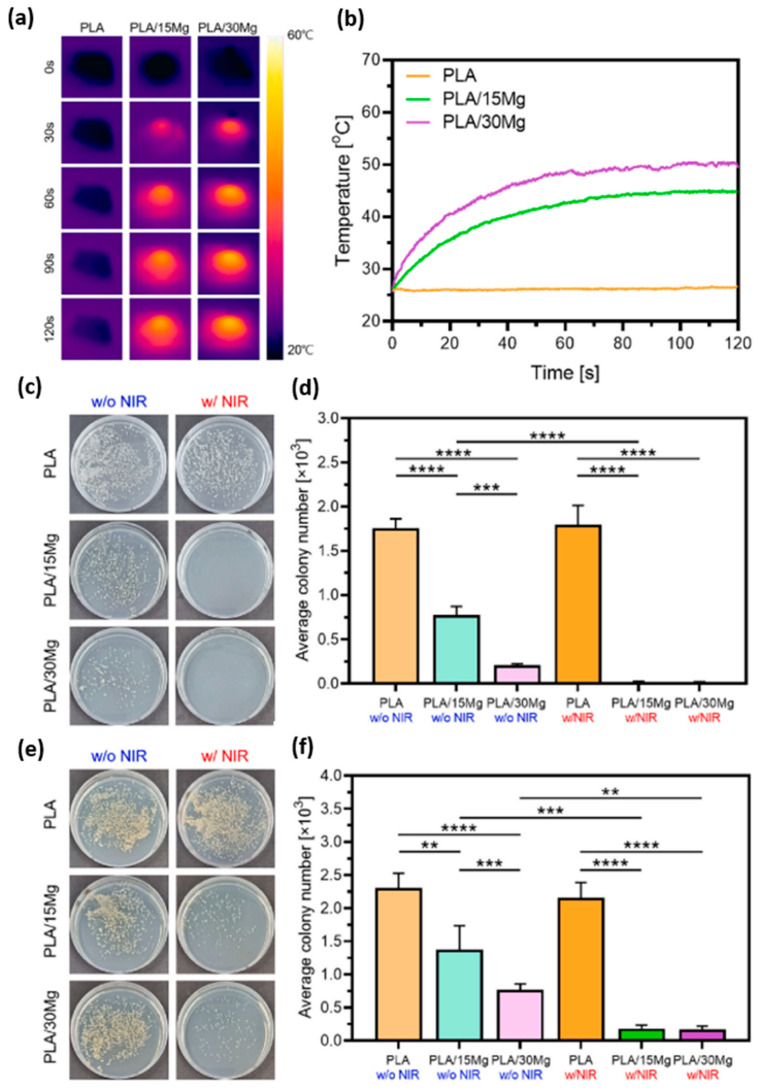
(**a**,**b**) Hyperthermia of magnesium ions under near infrared (NIR at 808 nm) application and (**c**–**f**) antibacterial capacity of magnesium ions under *E. coli* (**c**,**d**) and *S. aureus* (**e**,**f**); (** *p* < 0.01, *** *p* < 0.005, and **** *p* < 0.001). Adapted from [92]. This article was published in Biomater. Adv., 152 (May), 213,523, Lee, H et al. Antibacterial PLA/Mg Composite with Enhanced Mechanical and Biological Performance for Biodegradable Orthopedic Implants. Copyright, Elsevier, 2023.

**Figure 5 polymers-15-04667-f005:**
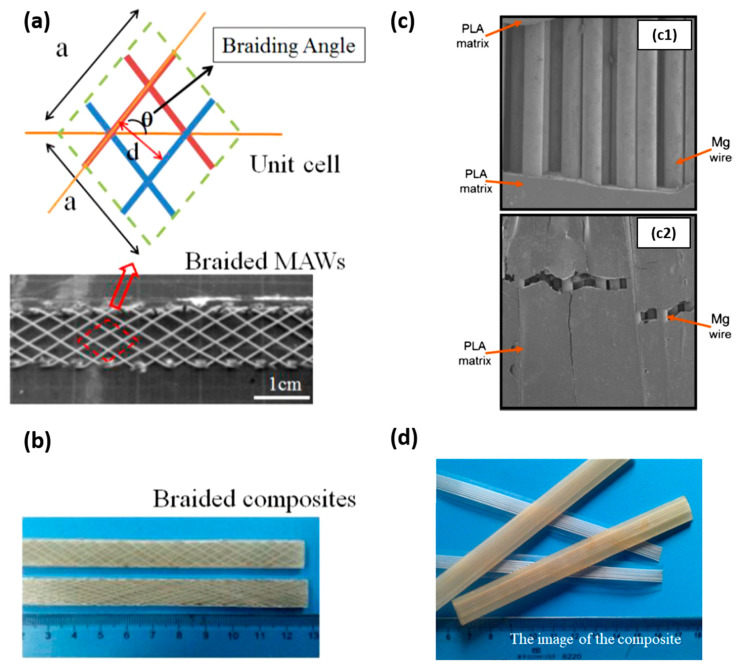
(**a**) Schematic illustration and structural features of braided magnesium wires; image adapted from [96], (**b**) image of braided PLA/Mg composites; image also adapted from [96], (**c**) SEM images of the fracture in tensile (**c1**) and flexural (**c2**) tests of unidirectional PLA/Mg composites; image adapted from [97]. (**d**) Image of unidirectional PLA/Mg composites; adapted from [98].

**Figure 6 polymers-15-04667-f006:**
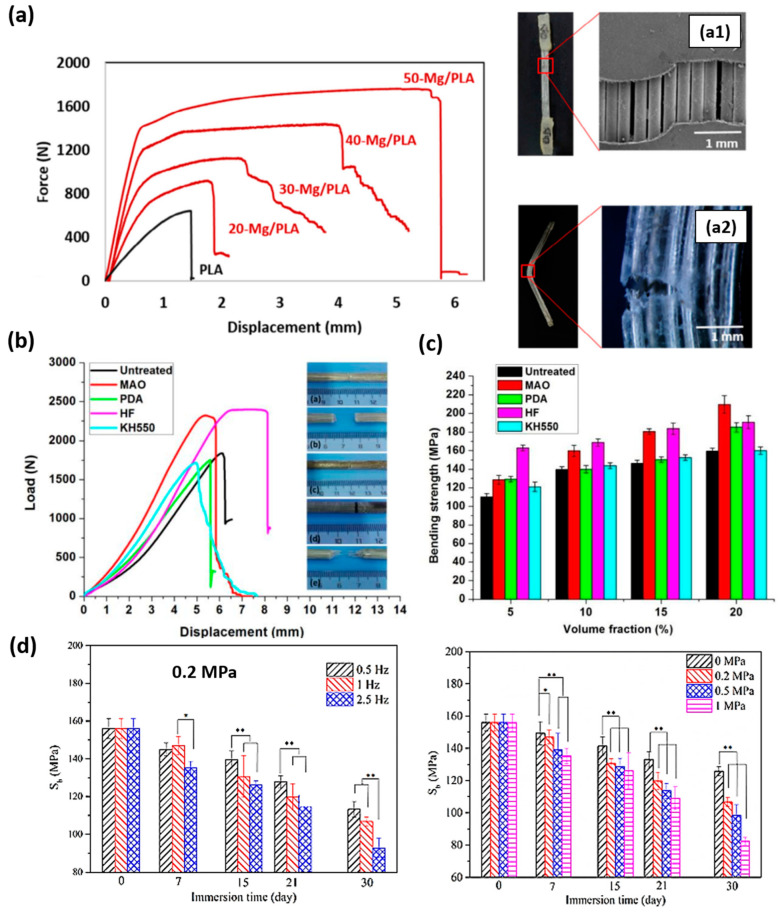
(**a**) Force displacement graphs from tensile tests of composites based on PLA reinforced with AZ31 wires with different volume fractions (**a**). Fracture morphologies after tensile (**a1**) and flexural tests (**a2**); image adapted from [99]. (**b**) Load displacement curves from tensile tests of composites based on PLA reinforced with Mg2Zn wires with different surface treatments (15 vol.% wire content) and fracture morphologies. (**c**) Bending strengths of PLA/Mg2Zn wires with different surface treatments as a function of wire content; image adapted from [100]. (**d**) Bending strength of composites based on PLA reinforced with AZ31 wires treated with MAO (20 vol% of wire content) as a function of immersion time under consistent dynamic conditions (0.2 MPa and different loading frequencies and 1 Hz and different predetermined loading magnitudes) with (* *p* < 0.05 and ** *p* < 0.01); image adapted from [98].

**Figure 7 polymers-15-04667-f007:**
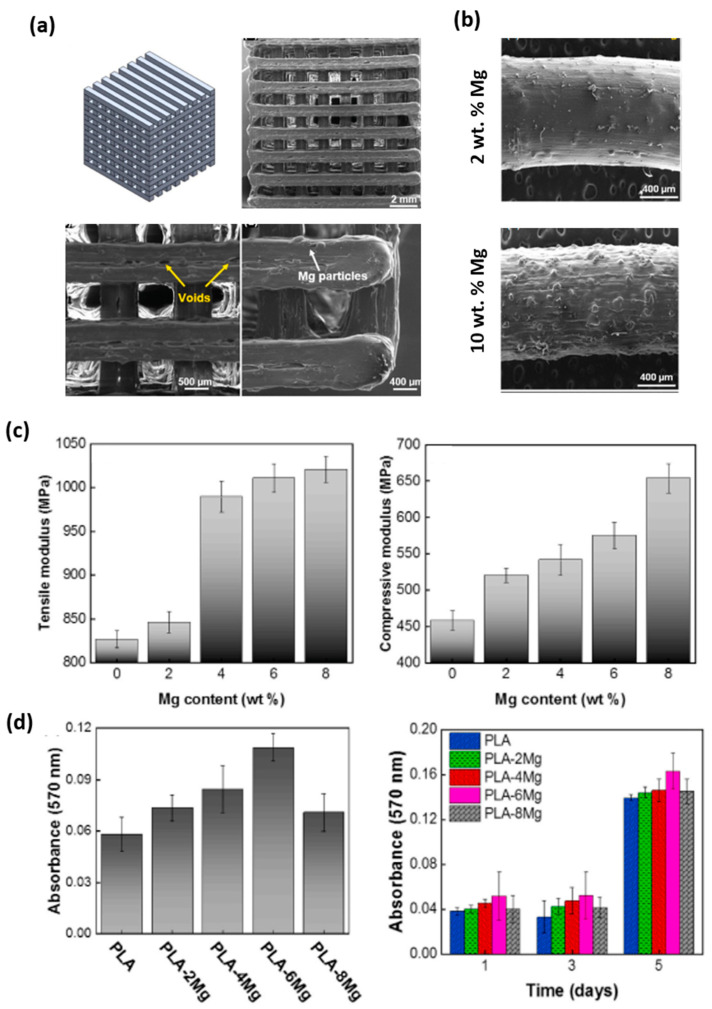
(**a**) Proposed scaffold design (porosity close to 47% and 800 µm pore size). (**b**) Longitudinal view of the filament and roughness associated with the increased presence of magnesium. (**c**) Tensile and compressive mechanical properties with increasing metal content. (**d**) Quantification of the L929 line in terms of cell adhesion and proliferation test. Images adapted from [108].

**Figure 8 polymers-15-04667-f008:**
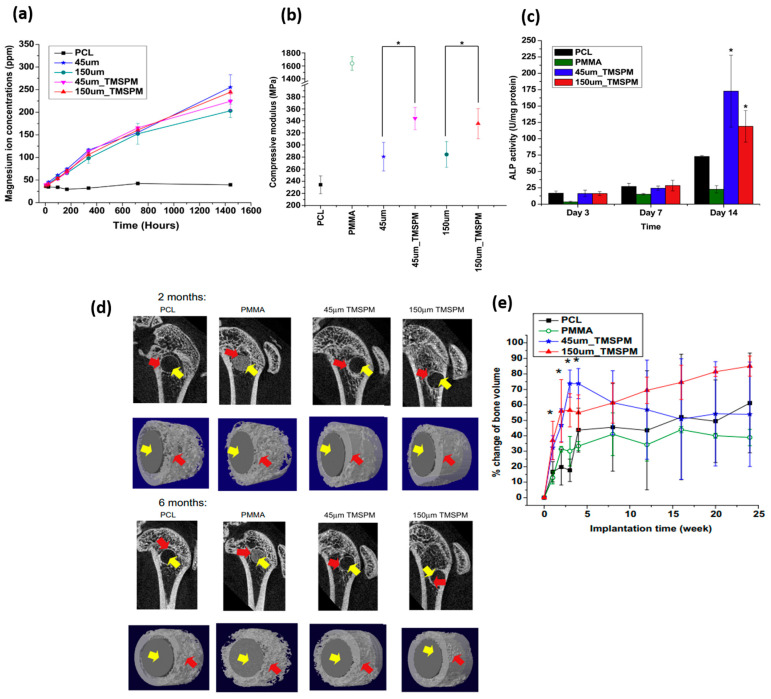
Images adapted from H.M. Wong et al. [116], where magnesium–silane reinforced PCL-based systems were analyzed. (**a**) Influence on the release of Mg^2+^ ions. (**b**) Elastic modulus values (MPa) in compression tests. (**c**) ALP analysis of magnesium-loaded samples in in vitro bone studies with MC3T3−E1 compared to control systems based on PMMA and PCL. (**d**) Images and analysis from micro-CT of the percentage of bone obtained also compared to control systems as a function of implantation time (yellow and red arrows are graft and new bone regenerated, respectively). (**e**) Quantified volume bone regeneration; difference significantly higher * (*p* < 0.05).

**Figure 9 polymers-15-04667-f009:**
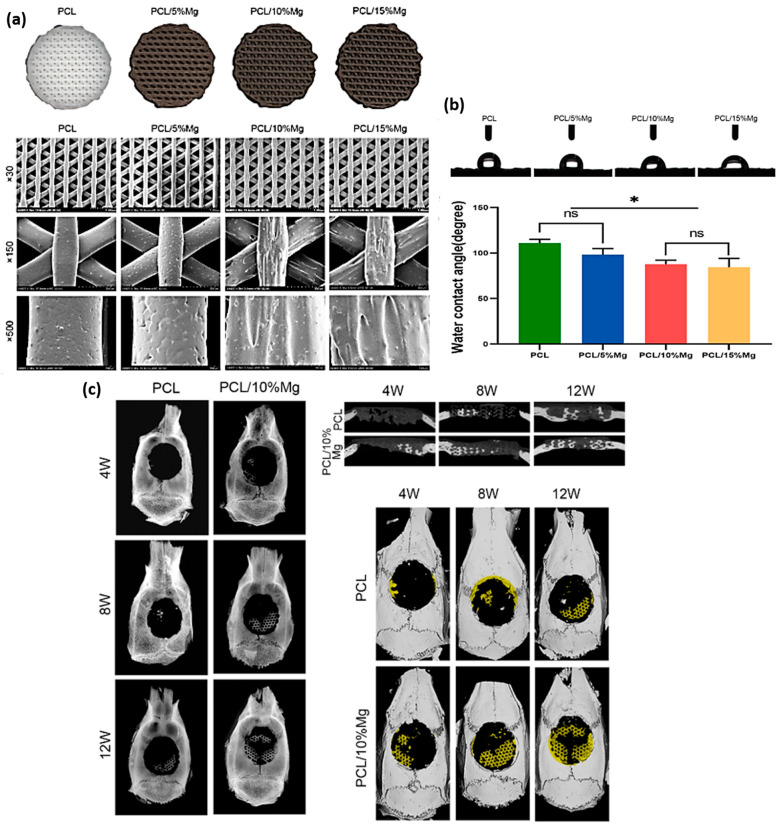
Images adapted from Zhao, S. et al. [125]. (**a**) Scaffold morphology observation of the 3D printed pure PCL and PCL/Mg scaffolds. (**b**) Measurements of water contact angles on the 3D printed pure PCL and PCL/Mg scaffolds; * *p* < 0.05. (**c**) X-ray images of SD rat calvaria defects implanted with porous scaffolds of pure PCL and PCL-10 wt.% at 4, 8 and 12 weeks; also, micro-CT cross-sectional images and 3D reconstruction images of SD rat calvarial defects implanted with porous scaffolds of pure PCL and PCL/10%Mg at 4, 8 and 12 weeks. Reprinted (adapted) with permission from ACS Biomater. Sci. Eng., 6 (9), 5120–5131. Zhao, S. et al., Fabrication and Biological Activity of 3D-Printed Polycaprolactone/Magnesium Porous Scaffolds for Critical Size Bone Defect Repair. Copyright, 2020, American Chemical Society.

**Table 3 polymers-15-04667-t003:** Mechanical properties of PLA-based composites reinforced with Mg wires before and after in vitro degradation.

Matrix	Mg Wires	Content (vol.%)	Surface Treatment	Tensile/Bending Strength (MPa)	Degradation Conditions	Degraded Bending Strength	Year, Reference
PLA (90,000–120,000 g/mol)	AZ31	0		59/-			2018, [99]
		20		85/131			
		30		109/177			
		40		135/196			
		50		164/245	PBS (35 days at 37 °C)	64	
					PBS (14 days at 50 °C)	42	
PLA (90,000–120,000 g/mol)	AZ31	50	MgF_2_	164/255	PBS (56 days at 37 °C)	108	2019, [97]
PLA (density 1.24 g/cm^3^)	AZ31	10	MAO	-/110	KBM (21 days at 37 °C) Dynamic tests (1 Hz)		2017, [101]
					0 MPa	90	
					0.1 MPa	82	
					0.3 MPa	75	
					0.9 MPa	65	
PLA (density 1.24 g/cm^3^)	AZ31	20	MAO	-/156	KBM (30 Days at 37 °C) Dynamic tests		2019, [98]
					1 Hz		
					0 MPa	128	
					0.2 MPa	110	
					0.5 MPa	100	
					1 MPa	80	
					0.2 MPa		
					0.5 Hz	112	
					1 Hz	108	
					2.5 Hz	90	
PLA (78,000 g/mol)	Mg2Zn	0		-/103	Hank’s solution (10 weeks at 37 °C)	80	2019, [102]
		5	MAO	-/129		20	
		5	MAO and hot drawn	-/207		25	
		10	MAO	-/160		36	
		10	MAO and hot drawn	-/211		45	

-: the authors did not evaluate this behavior.

**Table 4 polymers-15-04667-t004:** PLA/Mg particle composites, characteristics of the matrix, particle reinforcement and processing method.

PLA Matrix	Reinforcement (Size)/Surfaces Modification	Reinf. wt.% in Polymer	Processing Method	Year, Reference
PLLA (1.25 g/cm^3^)	Mg (<250 µm)/-	0 and 30	Solvent casting and compression molding	2012, [83]
PLLA (MFI: 35.8 g/10 min)	Mg (<50 µm)/-	0, 0.5, 1, 3, 5 and 7	Hot extrusion and compression molding	2017, [84]
PLLA (95,000 g/mol) and PDLLA (103,000 g/mol)	Mg (25 µm)/-	0, 1, 5, 10 and 15	Hot extrusion and compression molding	2017, [85]
PLA 2002D	Mg (25 µm)/-	0, 0.2 and 1	Injection molding	2016, [86]
PLA 2003D	Mg (29.22 µm)/PEI and CTAB	0, 5, 10, 30 and 50	Colloidal processing	2020, [88]
PLA 2002D	Mg (<50 µm) spherical and irregular particles/-	0 and 10	Hot extrusion and compression molding	2016, [89]
PLLA (89,000 g/mol) and PDLLA (95,000 g/mol)	Mg (25 µm) irregular particles/-	0 and 10	Hot extrusion and compression molding	2019, [90]
PLA (1.25 g/cm^3^)	Mg (100 µm)/-	0, 2 and 5	Solvent casting and compression molding	2017, [91]
PLA (4032D)	Mg (120 µm)/-	0, 15 and 30 vol.%	High shear processing	2023, [92]
PLA	Mg (<100 µm) irregular particles/-	0, 2, 4, 6, 8 and 10	FFF (3D-Printing)	2023, [108]
PLA (68,000 g/mol)	AZ61 (<100 µm)/-	0, 5, 10 and 15	FFF (3D-Printing)	2023, [111]
PLA pellets	MgO (<40 µm)/-	0, 10, 25 and 40	PBF-LB (3D-Printing)	2023, [113]

-: not applicable or not mentioned by the authors.

**Table 5 polymers-15-04667-t005:** Biological studies on PLA/Mg bioabsorbable composites with application in bone tissue engineering.

Sample	Preclinical Phase	Cell Line	Comments	Year/Reference
PDLLA with 0, 0.2 and 1 wt.% Mg	In vitro	hMSCs	Low presence of magnesium resulted in improved cell viability and macrophage responses	2016, [86]
PLLA and PDLLA with 10 wt.% Mg	In vitro	hMSCs	Mg favored cytocompatibility evaluation. On the other hand, the structures with higher amorphous content (PDLLA) showed better cellular response	2019, [90]
PLA with 0, 2 and 5 wt.% Mg	In vitro	MC3T3−E1	Improved cell viability	2017, [91]
PLA with 0, 15 and 30 vol.% Mg	In vitro	MC3T3−E1	Improved cell activity in terms of cell adhesion and proliferation compared to control and 30 vol.%. This later was associated with an increase in pH	2023, [92]
		Mg ion antibacterial properties with respect to *E. Coli* and *S. Aureus*	Antibacterial capacity of magnesium ions under near infrared (NIR) emission (808 nm)	
PLA (AZ31 wires with Ca-P)	In vitro	ADSCs	Mg improves cell adhesion and proliferation processes	2023, [103]
PLA with 0, 2, 4, 6, 8 and 10 wt.% Mg	In vitro	Human fibroblast (L929)	Up to 6% positive effect in terms of bone cell culture	2021, [108]
PLA with 0, 5, 10 and 15 wt.% (AZ61)	In vitro	MC7s epithelial line	Improved cell viability	2023, [111]

**Table 6 polymers-15-04667-t006:** PCL/Mg composites, characteristics of the matrix, reinforcement and processing method.

PCL Matrix (g/mol)	Reinforcement (Size)/Surface Modification	Reinforcement wt.% in Polymer	Processing Method	Year, Reference
80,000	Mg (45–150 µm)/TSPM	10	Blending at 60 °C	2013, [116]
80,000	Mg (45–150 µm)/TSPM	0, 5, 10 and 15	Salt leaching	2014, [118]
50,000	Mg(OH)_2_ (10 nm) and MgO (20 nm)	0, 0.5, 1, 5, 10 and 20	Electrospinning	2023, [119]
68,000	Mg (45 µm)/-	0, 5, 10 and 15	FFF (3D printing)	2020, [125]
80,000	Mg (28.6 µm)/-	0, 1, 3, 5, 7 and 9	FFF (3D printing)	2021, [128]
45,000	Mg(OH)_2_ (<50 nm)/-	0, 5 and 20	FFF (3D printing)	2020, [18]
-	Powder from ball milling mix [132]/-	0, 10, 20 and 30	LS (3D printing)	2017, [131]

-: not applicable or not mentioned by the authors.

**Table 7 polymers-15-04667-t007:** Biological studies on PCL/Mg bioabsorbable composites with application in bone tissue engineering.

Sample	Preclinical Phase	Cell line or Animal Model (Injury)	Comments	Year/Reference
PCL with 10 wt.% of Mg (treated with TMSPM)	In vitro	MC3T3−E1	ALP protein marker showed better differentiation towards bone lineage in the presence of Mg	2013, [116]
	In vivo	SD rats (lateral epicondyle)	Micro-CT analysis demonstrated increased bone formation in the presence of Mg	
PCL with 0, 5, 10 and 15 wt.% of Mg (treated with TMSPM)	In vitro	MC3T3−E1	High Mg levels favored stages such as cell adhesion and proliferation with respect to the control sample. In addition, ALP marker showed greater capacity for bone differentiation	2014, [118]
	In vivo	SD rats (lateral epicondyle)	Micro-CT analysis demonstrated increased bone formation with Mg presence	
PCL with 0, 5, 10 and 15 wt.% of Mg	In vitro	rBMSCs	Improvement of cellular processes up to 10 wt.%; from this percentage, a reduction in efficiency was observed due to an increase in the pH of the medium	2020, [125]
	In vivo	SD rats (skull model)	Micro-CT and X-ray measurements showed that the reinforced sample was significantly improved over the control sample	
PCL with 0, 1, 3, 5, 7 and 9 wt.% of Mg	In vitro	rBMSCs	Improvement of cellular processes such as cell adhesion and proliferation effective up to 5 wt.%. In addition, ALP marker showed that 3 wt.% exhibited better differentiation capacity towards bone lineage	2021, [128]
	In vivo	Rabbits (medial tibial tubercle)	Analysis of 3 wt.% Mg improved in terms of bone repair with respect to the control sample	
PCL with 0, 5 and 20 wt.% of Mg(OH)_2_	In vitro	hOBs	Improved cell adhesion and proliferation with the presence of Mg. On the other hand, ALP also showed a greater capacity for differentiation towards bone lineage	2020, [18]
PCL with 0, 10, 20 and 30 wt.% of CS-Mg	In vitro	hMSCs	Alizarin red staining and osteocalein showed higher bone matrix mineralization at 20 wt.% with respect to the control system. Synergy formed with the corresponding Mg and release of silicate ions	2017, [131]

## Data Availability

Data is available upon request.

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
