# Peer review of "Bioabsorbable Composites Based on Polymeric Matrix (PLA and PCL) Reinforced with Magnesium (Mg) for Use in Bone Regeneration Therapy: Physicochemical Properties and Biological Evaluation"

_polymers, 2023, doi:10.3390/polym15244667_

Round 1

Reviewer 1 Report

Comments and Suggestions for Authors

Evaluating the manuscript as a review and not a research article, I would like to note that it is a very good review effort in the field of bioabsorbable composites, focused on PLA and PCL polymeric matrices, reinforced with Mg; materials that were developed in the last 10-15 years.

The manuscript is characterized by completeness, clarity and it is well-structured.

The figures/tables/images/schemes have the appropriate captions and are easy understandable.

As for the cited references, they are mostly recent and relevant publications.

In conclusion, there is one notification: it is not clear in the introduction (lines 118-124) why the specific polymeric matrices, among others, were chosen for the study, although it is mentioned elsewhere that these polymers are two of the most studied.

Reviewer 2 Report

Comments and Suggestions for Authors

Dear Authors,

thank you for providing the review article regarding PLA and PCL-based biocomposites reinforced with Mg for bone tissue engineering. The review article is comprehensive and systematizes knowledge on the subject in a clean manner. The introduction is well outlined and revolves around TERM. Every subsequent element of the studies’ body is described thoroughly. 

Please address my comments enlisted below:

1. Abbreviation “TERM” should be explained in the abstract section as it is first part of the article (not in introduction)

2. Line 124: please remove “etc…” or at least triple dot.

3. Line 129: “certain animals” – please avoid generalisms – give an example of certain species. It is a general remarks for the whole article to avoid such structures.

4. Table 1: please include molecular weight (or its distribution) for PLA and its derivatives (data for commercial product is available).

5. I am lacking in discussion on influence of MW on bio absorption of polymer matrix – it is vital for implantation applications.

6. Since the main application of described material is bone tissue engineering I would expect a summary (preferably in the form of Table) of the most important in vivo and in vitro experiments reported in literature.

7. Please present the chemical structure of described polymers

After addressing comments presented above, this paper will be considered for publication by me.

Best regards,

Reviewer

Comments on the Quality of English Language

Proposed paper requires minor language editing.  
